# Life Table Parameters of the Tomato Leaf Miner *Tuta absoluta* (Lepidoptera: Gelechiidae) on Five Tomato Cultivars in China

**DOI:** 10.3390/insects15030208

**Published:** 2024-03-20

**Authors:** Hesen Yang, Chi Zhang, Yuyang Shen, Haifeng Gao, Guifen Zhang, Wanxue Liu, Hongbo Jiang, Yibo Zhang

**Affiliations:** 1College of Plant Protection, Southwest University, Chongqing 400715, China; 15532287670@163.com (H.Y.); jhb8342@swu.edu.cn (H.J.); 2State Key Laboratory for Biology of Plant Diseases and Insect Pests, Institute of Plant Protection, Chinese Academy of Agricultural Sciences, Beijing 100193, China; zhangguifen@caas.cn (G.Z.); liuwanxue@caas.cn (W.L.); 3Key Laboratory of Invasive Alien Species Control of Ministry of Agriculture and Rural Affairs, Beijing 100193, China; 4Rural Energy and Environment Agency, Ministry of Agriculture and Rural Affairs, Beijing 100125, China; zhangchimy19@163.com; 5Institute of Plant Protection, Xinjiang Academy of Agricultural Sciences, Urumqi 830091, China; sansirosoul@163.com (Y.S.); ghf20044666@163.com (H.G.)

**Keywords:** life table, duration of preadult stage, lifetime fecundity, adult longevity

## Abstract

**Simple Summary:**

As a newly invasive pest in China, the moth *Tuta absoluta* has spread extremely quickly, and now causes serious harm to the Chinese tomato industry. Understanding the resistance of *T. absoluta* to different commercially processed tomato cultivars can improve the integrated management strategy using resistant plants. In this study, we found four processed tomato cultivars (Th9, Th1902, Heinz1015, and Dimen2272) that have higher resistance to *T. absoluta* than fresh tomato (Dafen). Among the four processed tomato cultivars, Th9 was the most suitable cultivar, as *T. absoluta* showed the higher *r* value on Th9 than on the other three processed tomato cultivars.

**Abstract:**

Tomato is the most preferred host plant for *Tuta absoluta*, a newly emerged devastating invasive pest in China. However, no study has evaluated the damage risk of *T. absoluta* on processed tomato worldwide. In the current study, the life table parameters of *T. absoluta* were systematically investigated on five tomato cultivars (one fresh tomato cultivar, four processed tomato cultivars) to determine their susceptibility to *T. absoluta* infestation. *T. absoluta* had a better population growth ability on the fresh tomato, “Dafen”, showing shorter duration of the preadult stage, higher lifetime fecundity, and a higher intrinsic rate of increase compared to four processed tomato cultivars. Meanwhile, the life table parameters of *T. absoluta* among different processed tomato cultivars also showed significant differences. Th9 was the most susceptible to *T. absoluta* attack, while Th1902, Heinz1015, and Dimen2272 were the least suitable ones for its development and reproduction. In summary, these tomato cultivars are the most recommended for commercial tomato production to reduce the damage caused by *T. absoluta* and improve the integrated pest management strategy.

## 1. Introduction

The South American tomato leafminer *Tuta absoluta* (Meyrick) is native to Peru, western South America [1]. After invading Spain in 2006, it spread rapidly throughout Afro-Eurasia and has since become a major threat to the world’s tomato production [2,3,4,5]. *Tuta absoluta* has been considered among the most devastating pests of both open field and greenhouse tomatoes worldwide [6], which can attack all the aboveground plant parts and all the developmental stages (seedling and mature) of the tomato plant, resulting in considerable reduction in yield and quality of the fruits [1]. Its larvae mine and feed on leaves, stems, petioles, and even bore into the tomato fruits when the population density is high [4,5]. Once it has invaded a new country, *T. absoluta* can not only damage the tomato plants and fruits, directly resulting in economic costs, but it can also indirectly affect the international trade of tomato as the quarantine restrictions are established by importing countries [3,5].

Host range is constrained by the behavioral and physiological traits of the insect [7]. Since 2006, many researchers had paid attention to the biological performance and oviposition performance of *T. absoluta* on different plants (different *solanum* plants or different tomato cultivars) across the world, such as in South America [8,9,10,11], Europe [12,13,14,15], Asia [16,17,18,19,20,21,22], and Africa [23,24,25]. The general result was that tomato is the most preferred host plant for *T. absoluta* oviposition and is suitable for larval development [4]. However, some *solanum* plants or tomato cultivars can reduce the developmental capacity of the pest without requiring any technical skill on the part of the farmer, so it is a potential pathway to reduce chemical control of *T. absoluta* [4,6]. Resistance may be the result of the plant’s phytohormonal system, triggered when the plant is attacked by herbivores [26,27]. Meanwhile, the use of resistant or tolerant cultivars could be part of an integrated management strategy [28].

China is the largest tomato production region in the world. In 2021, China’s tomato planting area reached 1.11 million hectares, producing 67.63 million tons, accounting for 36% of the world’s total tomato production [29]. *Tuta absoluta* was first found in the Ili country of China in 2017 [30] and, after that, its population quickly expanded and was established in more than 10 provinces in China [31,32]. So far, based on the field investigation, *T. absoluta* could damage four cultured crops (tomato, eggplant, potato, and Chinese lantern) and two wild plants (black nightshade and Dutch eggplant) in China [31]. As a result, it has seriously threatened tomato production in China. Meanwhile, China is the world’s largest exporter of processed tomatoes, accounting for 23%, according to statistics from the World Processed Tomato Council in 2022. At present, the most widely planted processed tomato varieties in China are Heinz1015 (Heinzseed division of Kraft Heinz foods company, Stockton, Canada), Dimen2272 (Del Monte Foods company, Walnut Creek, CA, USA), Th9, and Th1902 (Cofco Tunhe Seed Company, Changji, China). However, there has been no study to assess the risk of *T. absoluta* damage to processed tomatoes in the world.

We hypothesized that different tomato cultivars have different genetic backgrounds and could show different resistant levels to *T. absoluta*. In the current study, the main commercial fresh and processed tomato cultivars available in China were collected, and the life table parameters of *T. absoluta* on different tomato cultivars were systematically investigated using the age-stage, two-sex life table method to determine their level of susceptibility to *T. absoluta* infestation.

## 2. Materials and Methods

### 2.1. Insects

The original population of *T. absoluta* was collected at Yangrui Organic Agriculture Base, Hongta District, Yuxi City, Yunnan Province, in December 2018. The insect larvae were reared on *Lycopersicon esculentum* (“Dafen” Tomato, Shandong Shouhe seed Industry Co. Ltd., Jinan, China) and kept in an artificial growth chamber (26 ± 1 °C, a photoperiod of 14 L:10 D, and relative humidity (RH) of 65 ± 10%) in the Institute of Plant Protection, Chinese Academy of Agricultural Sciences (IPPCAAS), Beijing.

### 2.2. Tomato Cultivars

This study selected four main processed tomatoes grown in Xinjiang as the host plants for the experiment, and used a type of fresh tomato, “Dafen”, raised with the laboratory *T. absoluta* population as a control. The processed tomato cultivars are “Th1902”, “Th9”, “Dimen2272”, and “Heinz1015”. All healthy plants were cultivated and provided in the greenhouse of IPPCAAS. These four cultivars of tomato were grown in a potting mix (a mixture of nutrient soil and vermiculite = 2:1) in a plastic square basin (10 × 10 × 10 cm). When the effective true leaves of all host plants grew to 5–7 leaves, they were used for the experiments. None of the tomato plants used in the experiment had been exposed to chemical pesticides.

### 2.3. Life Table Study

Five tomato cultivars (“Dafen”, “Th1902”, “Th9”, “Dimen2272”, and “Heinz1015”) were used as the host plant for *T. absoluta* and placed in a net cage (40 × 40 × 60 cm, each plant must have 5–7 or more real leaves). Between 20:00 and 22:00, 50 pairs of *T. absoluta* adults (preferably the newly emerged individual) were placed in each cage, allowing the female to lay eggs. After 4 h, *T. absoluta* were removed. The next day at 8:00, 100 eggs were randomly selected for each treatment. All eggs were put on fresh tomato leaves (one egg per leaf) and maintained in a plastic petri-dish (9 cm); then, they were incubated in an artificial climate chamber as previously described until incubation. The hatching status of the eggs and the survival of individual *T. absoluta* was observed daily at 20:00 under a dissecting microscope (Zeiss, Oberkochen, Germany). During the observations mentioned above, the humidity of the leaves was maintained, and the leaves were timely replaced to ensure an adequate food supply for the larvae. After pupation, the sex of the pupae was determined, and the pupal duration was recorded. After eclosion, the newly eclosed male and female adults were paired in a 1:1 ratio in a specially designed cylindrical plastic device (20 cm height, 10 cm diameter) and placed in a transparent plastic bowl filled with water at the bottom to moisturize the plants below. The base of the tomato plant branch with 3–5 leaves excised from the clean tomato plant was wrapped with cotton and placed in the plastic bowl filled with water, then the foliage portion was enclosed in the cylindrical plastic device described by Zhang et al. [32], in which newly eclosed *T. absoluta* adults were released. We provided each branch with fresh tomato leaves for oviposition and recorded the eclosion rate, the number of eggs, and developmental duration of adult *T. absoluta* until all individuals died. During this experiment, the replication of the treatments of *T. absoluta* on different host plants was 20. The life table studies of all cultivars were conducted under identical conditions and during the same time period.

### 2.4. Life Table Analysis

The life table raw data of *T. absoluta* were analyzed according to the age-stage, two-sex life table theory and the method described by Chi with the computer program TWOSEX-MSChart [33]. The age-stage life expectancy (*e_xj_*), reproductive value (*v_xj_*), and the intrinsic rate of increase (*r*) were calculated according to Chi and Liu [34].

The survival rate (*s_xj_*) (*x* = age and *j* = stage) is defined as the probability that a newly laid egg will survive to age *x* and stage *j*. The age-specific survival rate (*l_x_*) was calculated as follows:$$l_{x}=\overset{m}{\underset{j=1}{\sum }}s_{xj}$$
where *m* is the number of stages. The age-stage specific fecundity (*f_xj_*) and age-specific fecundity (*m_x_*) were calculated as follows:$$m_{x}=\frac{\sum_{j=1}^{m}s_{xj}f_{xj}}{\sum_{j=1}^{m}s_{xj}}$$

The net reproductive rate (*R*_0_) is defined as the total number of offspring that an individual can produce during its lifetime and was calculated as follows:$$R_{0}=\overset{\infty }{\underset{x=0}{\sum }}l_{x}m_{x}$$

The intrinsic rate of increase (*r*) was calculated using the Lotka–Euler equation with age indexed from 0, and was calculated as follows:$$\overset{\infty }{\underset{x=0}{\sum }}e^{-r\left(x+1\right)}l_{x}m_{x}=1$$

The finite rate (*λ*) was calculated as follows:$$\lambda =e^{r}$$

The mean generation time (*T*) represents the period that a population is required to increase to *R*_0_-fold of its size as time approaches infinity and the population settles down to a stable age-stage distribution. The mean generation time was calculated as follows:$$T=\frac{\ln R_{0}}{r}$$

### 2.5. Population Projection

To predict the population growth rate and the age-stage structure of 10 *T. absoluta* eggs over the next 60 days, the TIMING-MSChart program [35] was utilized. This prediction was based on multiple data inputs, including the hatching rate, survival rate, fecundity, and duration of each stage [34,36].

### 2.6. Statistical Analysis

The life history raw data of all *T. absoluta* individuals were analyzed based on the age-stage, two-sex life table using the TWOSEX-MSChart program [33]. The development duration and reproductive (oviposition and fecundity) parameters between different host plants were compared by using one-way ANOVA, complemented with the Tukey’s HSD method in SPSS statistics 26.0 version. Due to the non-normal distribution of the offspring sex ratio, the Kruskal–Wallis test was used for analysis. Moreover, according to Akca et al. [37], we used 100,000 re-samplings to obtain stable estimates of standard errors. A paired bootstrap test was used to detect statistical differences on *r*, *λ*, *R*_0_, *GRR*, and Doubling time of *T. absoluta* on different host plants. These procedures were embedded in the TWOSEX-MSChart program.

## 3. Results

### 3.1. Developmental Duration, Fecundity, and Offspring Sex Ratio of T. absoluta on the Five Tomato Cultivars

*Tuta absoluta* was able to complete its life-cycles when reared on different tomato cultivars. Tomato cultivar significantly affected the durations of the egg stage (F_4,225_ = 62.549, *p* < 0.05), larva stage (F_4,225_ = 225.203, *p* < 0.05), pupa stage (F_4,225_ = 10.375, *p* < 0.05), and preadult stage (F_4,225_ = 82.511, *p* < 0.05) of *T. absoluta*. For the egg stage, the longest duration was generated on Th9, and the shortest on Th1902. The durations of the larval stage and preadult stage of *T. absoluta* feeding on four cultivars of processed tomato were significantly longer than those feeding on Dafen. The longest durations of the larval stage and preadult stage of *T. absoluta* were generated on Th1902, respective up to 12.84 days and 22.74 days. The shortest durations of two stages of *T. absoluta* were generated on Dafen, up to 8.98 days and 19.14 days. Of the pupal stage, the longest pupal duration was seen on Heinz1015, and the shortest on Dimen2272.

The tomato cultivar also significantly affected the adult longevity (male: F_4,111_ = 8.036, *p* < 0.05; female: F_4,113_ = 8.655, *p* < 0.05), lifetime fecundity (F_4,113_ = 44.039, *p* < 0.05), and oviposition period (F_4,113_ = 15.239, *p* < 0.05), but did not change the offspring sex ratios (H_4,114_ = 3.962, *p* = 0.411). The shortest longevities of male and female adults were generated on Dimen2272, up to 9.83 days and 10.33 days, respectively. However, the longest longevities of male and female adults were seen on Dafen and Th1902 (Table 1). Similarly, the highest lifetime fecundity of *T. absoluta* was generated by Dafen (up to 172 eggs), while the lowest was seen on Dimen2272 (up to 44.88 eggs). The oviposition period of *T. absoluta* feeding on Th9, Th1902, and Heinz1015 was, respectively, 10.32 ± 0.74 days, 9.74 ± 0.87 days, and 8.10 ± 0.81 days, which were significantly longer than that of *T. absoluta* feeding on Dafen. In addition, *T. absoluta* reared on different processed tomato cultivars exhibited a difference in its fecundity. The total numbers of eggs per female significantly varied from 131.32 eggs on Th9, 81.74 eggs on Th1902, 73.95 eggs on Heinz1015, and 45.13 eggs on Dimen2272 (Table 1).

### 3.2. Life Table Analysis

Based on the age-stage specific survival rate (*S_xj_*) curves, the survival rate of *T. absoluta*, which feeds on five tomato cultivars from the egg to the pupa stage, maintained a high level, as seen in Figure 1. The lifespan of both male and female adults of *T. absoluta* feeding on Th1902 is longer than that of adults of *T. absoluta* feeding on other cultivars of tomato (Figure 1).

Moreover, there was an initial increase followed by a decrease in the female fecundity and age-specific reproduction of *T. absoluta* on all five tomato cultivars. The age-specific fecundity (*m_x_*) attained the reproductive high point at day 20, 23, 23, 22, and 21 on “Dafen”, “Th1902”, “Heinz1015”, “Th9”, and “Dimen2272”, respectively. The highest *m_x_* value recorded on “Dafen”, “Th1902”, “Heinz1015”, “Th9”, and “Dimen2272” was 23.5, 5.4, 5.2, 8.7, and 6, respectively. On “Dafen”, “Th9”, and “Dimen2272”, the *m_x_* curve was demonstrated to have only one peak. However, on “Th1902” and “Heinz1015”, more than one peak was observed in the *m_x_* curve, suggesting a difference in the oviposition period of the individuals. The female fecundity curves for *T. absoluta* on “Dafen”, “Th1902”, “Heinz1015”, “Th9”, and “Dimen2272” began on the 17th, 20th, 18th, 18th, and 19th days, respectively. The female fecundity parameters (*f_x_*) of *T. absoluta* feeding on “Th9” were higher compared to the other three processed tomato cultivars. However, the *f_x_* curve of *T. absoluta* feeding on four processed tomato cultivars was lower than that of *T. absoluta* feeding on “Dafen” (Figure 2).

The age-stage-specific reproductive values (*v_xj_*) of *T. absoluta* increased significantly when adults emerged (18 days on “Dafen”, 21 days on “Th1902”, 19 days on “Heinz1015”, 19 days on “Th9”, and 20 days on “Dimen2272”). The reproductive values of *T. absoluta* feeding on “Th9” were significantly higher compared to the other three processed tomato cultivars. Furthermore, as the lifespan of adults increases, reproductive values of *T. absoluta* feeding on “Dafen”, “Th1902”, “Heinz1015”, “Th9”, and “Dimen2272” gradually decreases to 0 on the 28th, 41st, 35th, 39th, and 31st days, respectively (Figure 3).

The age-stage life expectancy (*e_xj_*) refers to the amount of time an individual at ages *x* and stages *j* will continue to survive. Therefore, lifespan gradually decreases with age *x* until it reaches 0. The results revealed that the life expectancy at age 0 (*e*_01_) for *T. absoluta* feeding on “Dafen”, “Th1902”, “Heinz1015”, “Th9”, and “Dimen2272” was 32.77, 38.74, 35.85, 33.93, and 31.58 days, respectively (Figure 4).

The mean of net reproductive rate (*R*_0_), gross reproduction rate (*GRR*), intrinsic rate of increase (*r*), and finite rate of increase (*λ*) of the *T. absoluta* population feeding on “Th9” were found to be 66.24 ± 11.43, 71.43 ± 12.99, 0.1690 ± 0.0073, and 1.1531 ± 0.0095, respectively. These values were higher than those observed for *T. absoluta* feeding on the other three processed tomato cultivars. However, these values of *T. absoluta* feeding on four processed tomato cultivars were lower than those of *T. absoluta* feeding on “Dafen” (paired bootstrap test, *p* < 0.05). The mean generation time of the *T. absoluta* population feeding on “Dimen2272” was 23.74 ± 0.17, which was the lowest among the four processed tomato cultivars (paired bootstrap test, *p* < 0.05). Additionally, the population doubling time of *T. absoluta* feeding on “Th9” was 4.09 days, which was the lowest among all four processed tomato cultivars. However, the population doubling time of *T. absoluta* feeding on “Dafen” was only 3.41 days (Table 2).

### 3.3. Prediction of the Population Growth of T. absoluta on the Five Tomato Cultivars

The population growth dynamics of *T. absoluta* feeding on five tomato cultivars were observed over a period of 60 days using 10 eggs from each *T. absoluta* population. The results showed that the selected 10 eggs of *T. absoluta* feeding on “Dafen”, “Th1902”, “Heinz1015”, “Th9”, and “Dimen2272” could produce 90110.99, 14427.61, 13521.36, 42563.78, and 5059.66 offspring on the 60th day, respectively. The population of *T. absoluta* feeding on Th9 exhibited the fastest growth among all four processed tomato cultivars. The numbers of eggs, larvae, pupae, female adults, and male adults of *T. absoluta* feeding on Th9 on the 60th day were 6008.09, 29653.96, 6610.71, 100.08, and 190.94, respectively. After 60 days, the predicted population of *T. absoluta* feeding on Th9 was 2.95, 3.15, and 8.41 times higher compared to *T. absoluta* feeding on “Th1902”, “Heinz1015”, and “Dimen2272”, respectively (Figure 5).

## 4. Discussion

This is the first study investigating biological performance using a life table of the tomato pinworm on processed tomato plant species and fresh tomato plants. Our results indicated that *T. absoluta* can successfully complete its life-cycles on five tomato cultivars. Different tomato cultivars significantly affected the different developmental stages of *T. absoluta*. The shortest duration of the preadult stage generated on Dafen (up to 19 days), and the longest was seen on Th1902 (more than 22.74 days). These results were consistent with previous research. Rostami et al. [17] indicated that the durations of the preadult stage of *T. absoluta* generated significant differences among three tomato cultivars (Falkato, Grandella, and Isabella). Ghaderi et al. [18] found that tomato cultivars significantly affected the development of the preadult stage of *T. absoluta* compared with seven tomato cultivars in Iran. The longest duration of the preadult stage was found to be Early Urbana Y (up to 26.42 days), and the shortest was Cal JN3 (about 20.83 days). Meanwhile, Krechemer and Foerster [10] also found similar results by comparing the life history traits among six different tomato cultivars in Brazil in a quite low environmental temperature (20 ± 2 °C). In general, a host plant that provides the conditions for a faster development time will allow more generations of that pest to occur in the field. This means that this host plant could be the dominant plant for this pest, which is conducive to a rapid increase in the pest population. That is to say, it is more susceptible to damage from this pest. On the other hand, if a longer duration of the preadult stage was generated, it could increase the exposure to natural enemies and affect female individuals by forcing immature individuals to consume food resources of lower nutritional quality [38,39].

In the current study, *T. absoluta* produced the highest lifetime fecundity (up to 172 eggs) and the shortest female adult longevity (about 10.45 days) on Dafen, and showed the lowest lifetime fecundity on Dimen2272 (about 44.88 eggs) and the longest female adult longevity on Th1902 (up to 16.26 days). Similarly, these results also confirmed that Dafen, as a typical fresh tomato cultivar, is a more susceptible host plant than the rest of the four processed tomato cultivars. However, the lifetime fecundity of *T. absoluta* on Dafen was quite lower than the results in Younes et al. [25] and Heidari et al. [21]. By comparing with the detailed materials and methods of this research, we suggested that the reason for this difference could be that honey solution was absent in our experiment. It is well known that supplemental honey solution can not only increase the lifetime fecundity, but also prolong the adult longevity [40,41,42]. Moreover, among the four processed tomato cultivars, *T. absoluta* showed the highest lifetime fecundity on Th9 (up to 132 eggs), and the lowest on Dimen2272 (less than 45 eggs). It seems that the four processed tomato cultivars in our study showed quite different levels of resistances to *T. absoluta*.

To assess host plant resistance levels to insects and insects suitedness to host plants, the life table parameters, especially the intrinsic rate of natural increase (*r*), are the most important usable parameters. *r* is strongly influenced by the duration of time from birth to the start of reproduction and, to a lesser extent, by survival and reproduction rates [43,44]. Hence, it is a very useful index for evaluating the damage risk of a pest on different host plants. In this study, we found that the *r* value of *T. absoluta* was significantly affected by different tomato cultivars, and varied from 0.2034 d^−1^ on Dafen to 0.1310 d^−1^ on Dimen2272. Since *T. absoluta* had the shortest duration of the preadult stage, and the highest lifetime fecundity on Dafen, it is reasonable that the higher *r* value was generated on this cultivar. Conversely, the *r* values of *T. absoluta* on four processed tomato cultivars were significantly lower than on Dafen, which means that these processed tomato cultivars are unsuitable hosts and have higher resistance to *T. absoluta* compared to fresh tomato. Among the four processed tomato cultivars, Th9 was the most suitable cultivar as *T. absoluta* showed a higher *r* value on Th9 than on the other three processed tomato cultivars. This result was also reconfirmed by the expected population results, of which the predicted population of *T. absoluta* fed on Th9 after 60 days was 2.95, 3.15, and 8.41 times higher than that of *T. absoluta* fed on “Th1902”, “Heinz1015”, and “Dimen2272”, respectively.

Furthermore, the *r* values on Dafen (0.2034 d^−1^) and Th9 (0.1696 d^−1^) were significantly higher than some previous research. Ghaderi et al. [18] found that the *r* value of *T. absoluta* significantly changed among seven tomato cultivars in Iran, ranged from 0.1052 d^−1^ on tomato (Early Urbana Y) to 0.1522 d^−1^ on tomato (Cal JN3). Heidari et al. [21] indicated the *r* values of *T. absoluta* varied from 0.111 d^−1^ on tomato (Atrak) to 0.147 d^−1^ on tomato (King ston). Similar results were also found for six tomato cultivars in Brazil [10]. By conducting in-depth comparative analysis of these references, it was found that the main reason could be that *T. absoluta* was supplemented with honey water in these studies, which led to a significant increase in its adult longevity, as the tomato plants and experimental temperature in this research are similar. Insects have typically been able to acquire nutrients from randomly distributed non-host food resources such as floral and extra-floral nectar, homopteran honeydew, pollen, and leaf trichome exudates in the field [40,45,46]. Strictly speaking, however, these are random events with low probability, not inevitable events [47,48]. Therefore, it may not be very scientific to continuously provide honey solution to *T. absoluta* in indoor experiments, and may even cause the results of the experiment to deviate from the real results to some extent.

## 5. Conclusions

In conclusion, we systematically assessed the hazard risk of *T. absoluta* to five commercial tomato cultivars in China using the age-stage, two-sex life table method, including four processed tomato cultivars, for the first time. *Tuta absoluta* had better population growth ability on the fresh tomato, “Dafen”, showing shorter duration of the preadult stage, greater lifetime fecundity, and a higher intrinsic rate of increase compared to four processed tomato cultivars. Meanwhile, the life table parameters of *T. absoluta* among different processed tomato cultivars also showed significant differences. Th9 was the most susceptible to *T. absoluta* attack, while Th1902, Heinz1015, and Dimen2272 were the least suitable ones for its development and reproduction. Therefore, these tomato cultivars are the most recommended for commercial tomato production to reduce the damage caused by the tomato leafminer.

## Figures and Tables

**Figure 1 insects-15-00208-f001:**
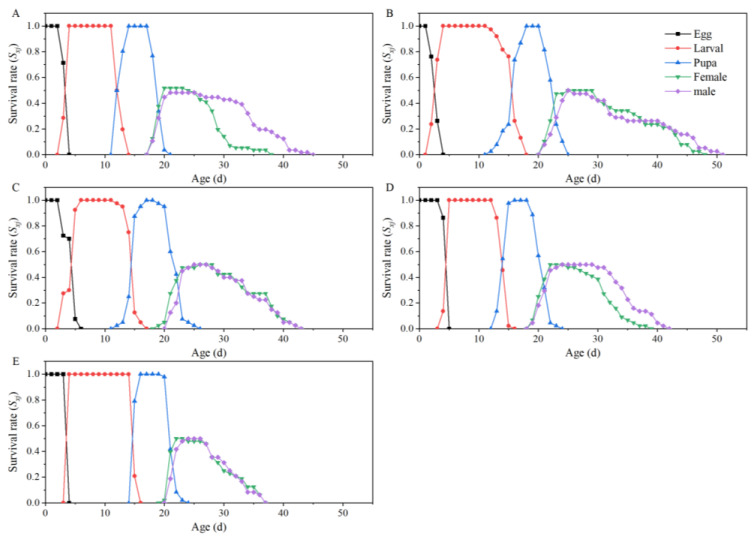
Survival rates at different developmental stages of *Tuta absoluta* feeding on different tomato cultivars. (**A**–**E**) The survival rates of *T. absoluta* after feeding on “Dafen”, “Th1902”, “Heinz1015”, “Th9”, and “Dimen2272”, respectively.

**Figure 2 insects-15-00208-f002:**
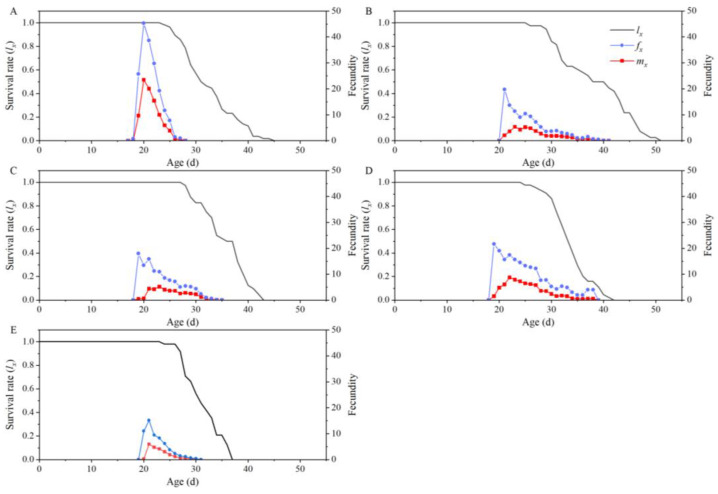
Age-specific survival rates, age-specific fecundity, and female fecundity of *Tuta absoluta* feeding on different tomato cultivars. (**A**–**E**) The reproduction rates and survival rates of *T. absoluta* feeding on “Dafen”, “Th1902”, “Heinz1015”, “Th9”, and “Dimen2272”, respectively. *l_x_*: Population survival rate; *m_x_*: population fecundity; *f_x_*: female fecundity.

**Figure 3 insects-15-00208-f003:**
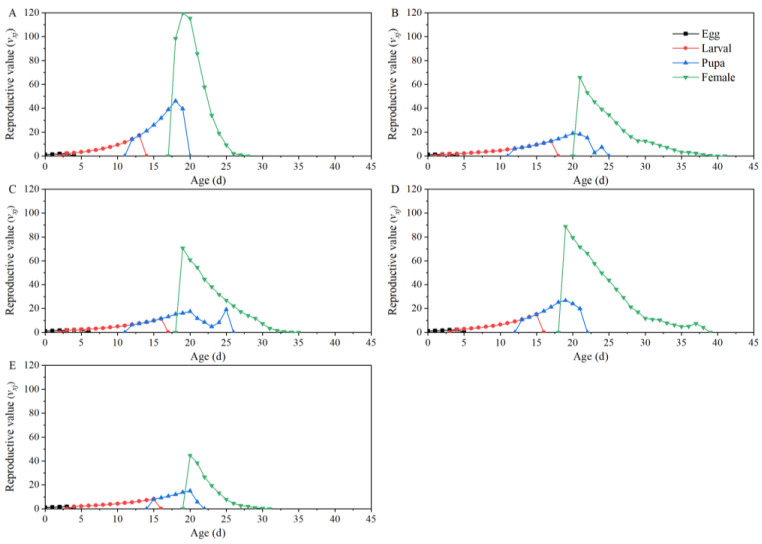
The reproductive value at each development stage of *Tuta absoluta* feeding on different tomato cultivars. (**A**–**E**) The life expectancy values for *T. absoluta* after feeding on “Dafen”, “Th1902”, “Heinz1015”, “Th9”, and “Dimen2272”, respectively.

**Figure 4 insects-15-00208-f004:**
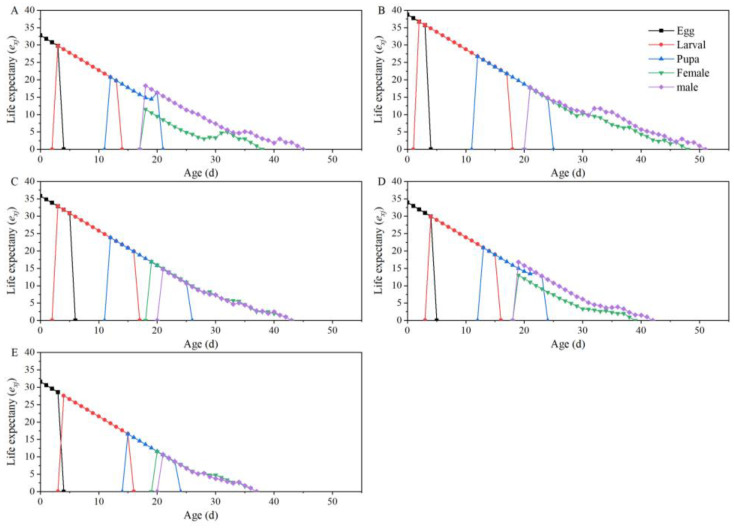
The life expectancy values at each development stage of *Tuta absoluta* feeding on different tomato cultivars. (**A**–**E**) The life expectancy values for *T. absoluta* after feeding on “Dafen”, “Th1902”, “Heinz1015”, “Th9”, and “Dimen2272”, respectively.

**Figure 5 insects-15-00208-f005:**
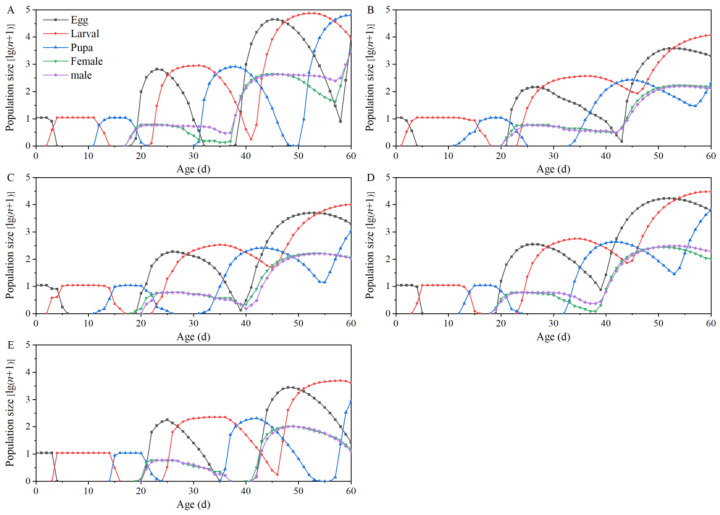
Population projection of *Tuta absoluta* feeding on five tomato cultivars by using the life tables of the original cohort with an initial population of 10 eggs. (**A**–**E**) The simulation of the population growth of *T. absoluta* after feeding on “Dafen”, “Th1902”, “Heinz1015”, “Th9”, and “Dimen2272”, respectively.

**Table 1 insects-15-00208-t001:** Durations of different developmental stages, lifetime fecundity, and offspring sex ratio (mean ± SE) of *Tuta absoluta* on five tomato cultivars.

Parameters	Dafen	Th1902	Heinz1015	Th9	Dimen2272
Duration of Egg stage	3.71 ± 0.06 c	3.03 ± 0.12 d	4.5 ± 0.16 a	4.86 ± 0.05 a	4 ± 0 c
Duration of Larva stage	8.98 ± 0.1 e	12.84 ± 0.13 a	10.35 ± 0.12 c	9.48 ± 0.08 d	11.21 ± 0.06 b
Duration of Pupa stage	6.45 ± 0.1 bc	6.87 ± 0.15 ab	7.25 ± 0.142 a	6.5 ± 0.11 bc	6.29 ± 0.08 c
Duration of preadult stage	19.14 ± 0.11 d	22.74 ± 0.20 a	22.10 ± 0.21 b	20.84 ± 0.17 c	21.50 ± 0.11 b
Adult male longevity	17.04 ± 0.86 a	15.74 ± 1.75 a	13.20 ± 1.02 ab	14.73 ± 0.72 a	9.83 ± 0.72 b
Adult female longevity	10.45 ± 0.56 c	16.26 ± 1.38 a	14.30 ± 0.98 ab	11.45 ± 0.70 bc	10.33 ± 0.80 c
Average oviposition period	5.00 ± 0.25 c	9.74 ± 0.87 a	8.10 ± 0.81 ab	10.32 ± 0.74 a	5.75 ± 0.51 bc
Lifetime Fecundity(eggs/female)	172.17 ± 7.50 a	80.89 ± 6.41 c	74.05 ± 9.24 cd	132.5 ± 11.22 b	44.88 ± 4.89 d
Offspring sex ratio (female/male)	1.07 ± 0.06 a	0.92 ± 0.05 a	1.01 ± 0.07 a	0.98 ± 0.04 a	1.10 ± 0.10 a

One-way ANOVA: Tukey’s HSD. This was used to detect the differences between different hosts. Significant differences between different treatments of the same parameter are indicated by a, b, c, d, and e (*p* < 0.05).

**Table 2 insects-15-00208-t002:** The population parameters (mean ± SE) of *Tuta absoluta* on five tomato cultivars.

Parameters	Dafen	Th1902	Heinz1015	Th9	Dimen2272
Net reproductive rate (*R*_0_)	89.16 ± 12.10 a	40.45 ± 7.24 bc	37.03 ± 7.40 cd	66.25 ± 11.43 ab	22.44 ± 4.02 d
Gross reproduction rate (*GRR*)	89.54 ± 12.17 a	44.10 ± 8.16 bc	38.25 ± 7.65 cd	71.43 ± 12.99 ab	22.98 ± 4.16 d
Intrinsic rate of increase (*r*, per day)	0.2034 ± 0.0063 a	0.1404 ± 0.0073 c	0.1432 ± 0.0082 c	0.1696 ± 0.0073 b	0.1310 ± 0.0076 c
Finite rate of increase (*λ*, per day)	1.2256 ± 0.0077 a	1.1507 ± 0.0084 c	1.1539 ± 0.0095 c	1.1848 ± 0.0086 b	1.1400 ± 0.0086 c
Mean generation time (days)	22.07 ± 0.17 d	26.36 ± 0.44 a	25.22 ± 0.34 b	24.73 ± 0.26 b	23.74 ± 0.17 c
Doubling time (days)	3.41 ± 0.11 c	4.94 ± 0.27 a	4.84 ± 0.29 a	4.09 ± 0.18 b	5.29 ± 0.32 a

One-way ANOVA: Tukey’s HSD. This was used to detect the differences between different hosts. Significant differences between different treatments of the same parameter are indicated by a, b, c and d (*p* < 0.05).

## Data Availability

The data presented in this study are available on request from the corresponding author.

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
