# Peer review of "Life Table Parameters of the Tomato Leaf Miner Tuta absoluta (Lepidoptera: Gelechiidae) on Five Tomato Cultivars in China"

_insects, 2024, doi:10.3390/insects15030208_

Round 1

Reviewer 1 Report (Previous Reviewer 1)

Comments and Suggestions for Authors

This study found four processed tomato cultivars (Th9,21Th1902, Heinz1015 and Dimen2272) have higher resistance to T. absoluta than fresh tomato (Dafen). Among the four processed tomato cultivars, Th9 was the most suitable cultivar as T. absoluta showed the higher value on Th9 than on the other three processed tomato cultivars. This provides a reference for select resistant tomato varieties to control T. absoluta. The areas that need to be revised are as follows:

1. In line 68 , one less of the five harmful crops is listed.

2. In line 100, “are” revised as “were

Author Response

Point 1: In line 68, one less of the five harmful crops is listed.

Response: Thanks for your suggestions. Corrected.

Point 2: In line 100, “are” revised as “were”

Response: Thanks for your suggestion. Per your comments, we have corrected the word.

Reviewer 2 Report (New Reviewer)

Comments and Suggestions for Authors

The study appears to have been carefully conceived, appropriately carried out, and well-written, analyzed and interpreted.

This contributes to an area of major interest and concern world wide-that of understanding crop/pest/pest control with a view to providing increased sustainable and environmentally yet effective pest control. Rather than examining pest control methods, here the susceptibility of the crop itself is examined. Do tomato cultivars differ in susceptibility to tomato leaf miner Tuta absoluta and is this a possible pathway to reduced chemical control of an important pest? More information in the introduction regarding the current situation would be of interest- are all studied cultivars grown extensively? Are there observed differences in T. absoluta infestation/pest pressure/crop losses? And chemical application? If so please add this information. Is there real data from where these cultivars are grown? Regarding response to geographic variables such as temperature/rainfall and pest damage or population of T. absoluta observed?

Indeed results show differences in susceptibility and the authors recommend cultivars for commercial tomato production with a view to reducing pest damage, reducing reliance on chemical control and improve opportunity for IPM.

Will length of larval stage impact on potential damage? Caterpillars around for longer?

Author Response

Response to Reviewer 1

The study appears to have been carefully conceived, appropriately carried out, and well-written, analyzed and interpreted.

This contributes to an area of major interest and concern world wide-that of understanding crop/pest/pest control with a view to providing increased sustainable and environmentally yet effective pest control. Rather than examining pest control methods, here the susceptibility of the crop itself is examined.

Point 1: Do tomato cultivars differ in susceptibility to tomato leaf miner Tuta absoluta and is this a possible pathway to reduced chemical control of an important pest?

Response: Thanks. Based on the results of previous research (line 50-58), it is certain that different tomato cultivars may have quite different susceptibility to Tuta absoluta, so resistant tomato cultivars could be a possible way to reduce chemical control of this pest. We have added this content in Line57-58.

Point 2: More information in the introduction regarding the current situation would be of interest- are all studied cultivars grown extensively?

Response: thanks, we added this contents in Line 73-76.

Point 3: Are there observed differences in T. absoluta infestation/pest pressure/crop losses? And chemical application? If so please add this information.

Response: Thanks, this is a very good comment. In the present study, we only investigated the life table parameters of T. absoluta on five tomato cultivars to determine their level of susceptibility to T. absoluta infestation indoors. In fact, we have not observed the differences in T. absoluta pest pressure/crop losses so far. Therefore, we will continue to improve these studies in the future. Besides, regarding chemical application, we did not use any chemical application in the present study and added the contents in Line 101-102.

Point 4: Is there real data from where these cultivars are grown? Regarding response to geographic variables such as temperature/rainfall and pest damage or population of T. absoluta observed?

Response: Thanks, this is a very good comment. Similar to the last comment. In the present study, we investigated the life table parameters of T. absoluta on five tomato cultivars indoor. To be indoor research, the data we collected in the present study could not be the real data from where these cultivars are grown. However, we will further conduct some field experiment to improve these studies in the future, and can try to consider geographic variables (temperature/rainfall/pest damage).

Point 5: Will length of larval stage impact on potential damage? Caterpillars around for longer?

Response: Yes, the length of the larval stage can affect potential damage. In general, for phytophagous insects, the rate of individual development could be slowed and the developmental duration will be prolonged if an unappetizing or resistant host plant or food are provided. When the length of larval or pupal stage is prolonged, the life history traits (adult fecundity, longevity) and life table parameter (net reproductive rate, intrinsic rate of increase, etc.) will be affected to some extent, so can further impact on potential damage.

This manuscript is a resubmission of an earlier submission. The following is a list of the peer review reports and author responses from that submission.

Round 1

Reviewer 1 Report

Comments and Suggestions for Authors

This study found four processed tomato cultivars (Th9,21Th1902, Heinz1015 and Dimen2272) have higher resistance to T. absoluta than fresh tomato (Dafen). 22 Among the four processed tomato cultivars, Th9 was the most suitable cultivar as T. absoluta showed 23the higher r value on Th9 than on the other three processed tomato cultivars. This provides a reference for select resistant tomato varieties to control T. absoluta. The areas that need to be revised are as follows:

1. In line 19, “comercial” revised as “commercial”

2. In line 22, “and” add before “,”

3. In line 42, world” revised as world’s”

4. In line 69, “and” add before “,”

5. In line 100, adult” revised as “adults

6. In line 103, “chamberas” revised as chamber as

7. In line 152, “non normal” revised as “non-normal”

8. In line 162, “significant” revised as “significantly

9. In line 164, generated” revised as “was generated

10. In line 171, “significant” revised as “significantly

11. In materials and methods, only the description of the treatment, how to design the repetition in the experiment ?

12. In the introduction, there is no clear research significance. the importance of this study shoud to be reorganized 

Comments on the Quality of English Language

Minor editing of English language required

Author Response

This study found four processed tomato cultivars (Th9,21Th1902, Heinz1015 and Dimen2272) have higher resistance to T. absoluta than fresh tomato (Dafen). 22 Among the four processed tomato cultivars, Th9 was the most suitable cultivar as T. absoluta showed 23the higher r value on Th9 than on the other three processed tomato cultivars. This provides a reference for select resistant tomato varieties to control T. absoluta. The areas that need to be revised are as follows:

  1. In line 19, “comercial” revised as “commercial”

Response: revised

  1. In line 22, “and” add before “,”

Response: revised

  1. In line 42, “world” revised as “world’s”

Response: revised

  1. In line 69, “and” add before “,”

Response: revised

  1. In line 100, “adult” revised as “adults”

Response: revised

  1. In line 103, “chamberas” revised as “chamber as”

Response: revised

  1. In line 152, “non normal” revised as “non-normal”

Response: revised

  1. In line 162, “significant” revised as “significantly”

Response: revised

  1. In line 164, “generated” revised as “was generated”

Response: revised

  1. In line 171, “significant” revised as “significantly”

Response: revised

  1. In materials and methods, only the description of the treatment, how to design the repetition in the experiment ?

Response: revised at Line 124-125

  1. In the introduction, there is no clear research significance. the importance of this study shoud to be reorganized 

Response: revised at line 78-79

Reviewer 2 Report

Comments and Suggestions for Authors

The paper under review is dedicated to a promising and interesting topic, namely insect pest fitness on different host plant cultivars. This direction of research is mainstream in modern entomology and plant immunity, being extensively developed by research groups all over the world. Moreover, Tuta absoluta is a notorious pest worldwide which recently invaded many new areas of at least three continents and its expansion goes on. The agricultural production of tomatoes and other crops is therefore under a threat and detailed studies are needed to predict and prevent the pest invasion. Several works have been carried out in different countries concerning local variability of resistance of tomato cultivars to T. absoluta. The particular goal of the reviewed manuscript is a comparison of fresh and processed tomatoes suitability as host plant of T. absoluta. This could be a sound approach with broad prospectives if at least 10-20 cultivars were provided for the comparison in each group. However, when the study is based upon one fresh and four processed cultivars, it makes the comparison senseless.

The data themselves may be of some interest to researchers focused on regional cultivar differences or local producers, but not to the broad audience of the Journal. Anyway, the manuscript suffers not only from disbalanced conclusions not supported by a sufficient dataset, but also from poor style and grammar, unclear presentation of illustrative material, etc.

I therefore recommend to reject the paper as the whole research framework must be drastically improved to recruit a sound dataset.

Comments on the Quality of English Language

Poor grammar and style

Author Response

In the first part, it is unbelievable that this reviewer chose “not applicable” for the following six questions.

The first question is “Does the introduction provide sufficient background and include all relevant references?”.

Response: Line 54-65, we have summarized all the literature on host fitness and/or life table of Tuta absoluta on different host plants in the last two decades, providing sufficient background and including and citing almost all relevant references.

The second question is “Are all the cited references relevant to the research?”

Response: Same reply as the previous one

The third question is “Is the research design appropriate?”

Response: In this study, we collected five major commercial fresh and processed tomato cultivars and investigated the life table parameters of T. absoluta on different tomato cultivars using the age-stage, two-sex life table method to determine their susceptibility to T. absoluta infestation. In my opinion, our research design was quite appropriate. I do not know why this reviewer chose “not applicable”, as he/she did not provide any detailed comments on our manuscript.

The fourth question is “Are the methods adequately described?”

Response: Same reply as the previous one.

The fifth question is “Are the results clearly presented?”

The sixth question is “Are the conclusions supported by the results?”

Response: The age-stage, two-sex life table method is the most typical and widely used analytical method for life table parameters. We have prepared and written the results and discussion part very carefully. We dare not say that there is no improvement in these two parts, but at least it cannot be not applicable.

The paper under review is dedicated to a promising and interesting topic, namely insect pest fitness on different host plant cultivars. This direction of research is mainstream in modern entomology and plant immunity, being extensively developed by research groups all over the world. Moreover, Tuta absoluta is a notorious pest worldwide which recently invaded many new areas of at least three continents and its expansion goes on. The agricultural production of tomatoes and other crops is therefore under a threat and detailed studies are needed to predict and prevent the pest invasion. Several works have been carried out in different countries concerning local variability of resistance of tomato cultivars to T. absoluta. The particular goal of the reviewed manuscript is a comparison of fresh and processed tomatoes suitability as host plant of T. absoluta.

This could be a sound approach with broad prospectives if at least 10-20 cultivars were provided for the comparison in each group. However, when the study is based upon one fresh and four processed cultivars, it makes the comparison senseless. 

Response: We would like to say that so far, no study has been able to assess the life table of an insect on 10-20 different host plants using the age-stage, two-sex life table method, because this kind of study has a huge workload and is very complicated. Generally, most of the studies conducted life table research on 3-5 different host plants, such as Kanle Satishchandra et al. (2019), Heidari et al. (2020). If someone suggests that we do this research on 10-20 host plants, we have to say that they might be crazy or have malicious intentions.

Kanle Satishchandra, N.; Chakravarthy, A. K.; Özgökçe, M. S.; Atlihan, R. Population growth potential of Tuta absoluta (Meyrick) (Lepidoptera: Gelechiidae) on tomato, potato, and eggplant. J. Appl. Entomol. 2019, 143 (5), 518–526. https://doi.org/10.1111/jen.12622.

Heidari, N.; Sedaratian-Jahromi, A.; Ghane-Jahromi, M.; Zalucki, M. P. How bottom-up effects of different tomato cultivars affect population responses of Tuta absoluta (Lep.: Gelechiidae): a case study on host plant resistance. Arthropod-Plant Interact. 2020, 14 (2), 181–192. https://doi.org/10.1007/s11829-020-09739-8.

The data themselves may be of some interest to researchers focused on regional cultivar differences or local producers, but not to the broad audience of the Journal.

Response: Before conducting this research, we carefully summarized all the literatures on host fitness and/or life table of Tuta absoluta on different host plants in the last two decades. Based on this, we make sure that there was no study to evaluate the life table parameters of T. absoluta on processed tomato cultivars worldwide, so our study could be the first time to evaluate the life table of T. absoluta on processed tomato cultivars and has certain implications. At least, we think this study is very suitable to Insects.

Anyway, the manuscript suffers not only from disbalanced conclusions not supported by a sufficient dataset, but also from poor style and grammar, unclear presentation of illustrative material, etc.

Response: We sincerely welcome suggestions from any reviewer, but we do not accept suggestions from rude reviewers who completely disagree with our research without offering any specific suggestions for change.